# Molecular characterization of antimicrobial resistance related genes in *E. coli, Salmonella* and *Klebsiella* isolates from broilers in the West Region of Cameroon

**Jude Fonbah Leinyuy**[1], **Innocent Mbulli Ali**[1,2]*, **Karimo Ousenu**[1], **Christopher B. Tume**[1,3]

1 Research Unit of Microbiology and Antimicrobial Substances, Department of Biochemistry, University of Dschang, Dschang, Cameroon, 2 The Biotechnology Centre, University of Yaoundé 1, Yaoundé, Cameroon, 3 Department of Biochemistry, Faculty of Science, University of Bamenda, Bamenda, Cameroon

* innocent.mbulli@univ-dschang.org

**Data Availability Statement:** All relevant data are within the paper and its Supporting Information files.

## Abstract

### Background

Antibiotic resistance has become an enduring threat to human health. This has prompted extensive research to identify the determinants responsible in a bid to fight the spread of resistance and also develop new antibiotics. However, routine procedures focus on identifying genetic determinants of resistance only on phenotypically resistant isolates. We aimed to characterise plasmid mediated resistance determinants in key *Enterobacteriaceae* isolates with differential phenotypic susceptibility profiles and evaluated the contribution of resistance genes on phenotypic expression of susceptibility.

### Methods

The study was carried out on 200 *Enterobacteriaceae* isolates belonging to the genera *E. coli*, *Salmonella*, and *Klebsiella*; 100 resistant and 100 susceptible to quinolones, aminoglycosides, and ESBL-producing as determined by disk diffusion. Reduced susceptibility in susceptible isolates was determined as an increased MIC by broth microdilution. Plasmid-borne resistance genes were sought in all isolates by endpoint PCR. We performed correlations tests to determine the relationship between the occurrence of resistance genes and increased MIC in susceptible isolates. We then used the notion of penetrance to show adequacy between resistance gene carriage and phenotypic resistance as well as diagnostic odds ratio to evaluate how predictable phenotypic susceptibility profile could determine the presence of resistant genes in the isolates.

### Results

Reduced susceptibility was detected in 30% (9/30) ESBL negative, 50% (20/40) quinolone-susceptible and 53.33% (16/30) aminoglycoside-susceptible isolates. Plasmid-borne resistance genes were detected in 50% (15/30) of ESBL negative, 65% (26/40) quinolone

**Funding:** The authors received no specific funding for this work.

**Competing interests:** The authors have declared that no competing interests exist.

susceptible and 66.67% (20/30) aminoglycoside susceptible isolates. Reduced susceptibility increased the risk of susceptible isolates carrying resistance genes (ORs 4.125, 8.36, and 8.89 respectively for ESBL, quinolone, and aminoglycoside resistance genes). Resistance gene carriage correlated significantly to reduced susceptibility for quinolone and aminoglycoside resistance genes (0.002 and 0.015 at $CI_{95}$). Gene carriage correlated with phenotypic resistance at an estimated 64.28% for ESBL, 56.90% for quinolone, and 58.33% for aminoglycoside resistance genes.

## Conclusions

A high carriage of plasmid-mediated genes for ESBL, quinolone, and aminoglycoside resistance was found among the *Enterobacteriaceae* tested. However, gene carriage was not always correlated with phenotypic expression. This allows us to suggest that assessing genetic determinants of resistance should not be based on AST profile only. Further studies, including assessing the role of chromosomal determinants will shed light on other factors that undermine antimicrobial susceptibility locally.

## 1 Introduction

*In-vitro* antibiotic susceptibility testing (AST) aims at predicting if an antimicrobial agent (AMA) will kill an infectious organism *in-vivo* in the infected host if used in therapy. This is done to solve the problem arising when empirical treatment fails due to the development of antimicrobial resistance (AMR). AST is thus primarily of clinical importance because it provides information that guides clinical decision making in selecting the appropriate AMA. Furthermore, it provides baseline data for guidelines on first-line empiric therapy in a given region, given the public health approach to the treatment of bacterial pathogens in resource-limited settings. Lastly, it supports epidemiological surveillance of the emergence of antibiotic resistance, resistance determinants, and their dissemination [1–3]. There are three *in-vitro* AST methods that are known to consistently provide reproducible and repeatable results: the disk diffusion, broth dilution, and agar dilution. These methods, though very helpful and adequate for clinical and epidemiological practices, do not provide direct information about molecular determinants responsible for the observed resistance [2, 4]. Molecular methods cover this gap by seeking for the genes responsible for the expressed resistance and the relative implications of these genes in the resistance phenomenon. However, phenotypic resistance may also be caused by other diverse mechanisms whose determination is often complex for routine procedures, laborious, expensive, and lacks standardization [5]. Mechanisms with molecular bases and epidemiological importance, such as resistance conferred by genes borne on mobile genetic elements (MGEs) or due to chromosomal recombinations induced by stress, such as antibiotic misuse, are determined only by molecular means [6, 7]. Some of the common molecular tools for the determination of AMR include polymerase chain reaction (PCR), DNA microarray, sequencing, and *in-silico* simulations and analyses [8, 9]. *In-vitro* AST gives results as susceptible, intermediate or resistant based on a threshold concentration of AMA, the minimum inhibitory concentration (MIC). These results, given as such, serve the purpose of AST in a clinical setting but provide no information on the genes responsible. Routine procedure in the search for AMR determinants involves selection of resistant strains by *in-vitro* AST followed by identification of resistance genes or determinants (e.g. mutations) by

molecular methods [10–12]. With AMR being a polygenic inheritance, it is difficult to conclude that the presence of one resistance determinant–a given resistance gene, for example–is solely responsible for the expressed resistance. On the other hand, does the presence of one resistance determinant–a given resistance gene–determine the expression of the resistance phenotype? Does phenotypic susceptibility give an idea of the level of expression of a resistance determinant? We focused on 3 genera of *Enterobacteriaceae* isolated from broilers including *Escherichia coli*, *Salmonella* spp. and *Klebsiella* spp., well known for their extensive colonization of guts of mammals and birds as commensals or pathogens [13, 14]. This in a bid to establish whether there was a clear-cut idea that susceptibility equals absence of resistance genes or determinants and also the molecular basis for reduced susceptibility in *in vitro* AST.

## 2 Methods

### 2.1 Study site

The West Region is one of the 10 Regions of the Republic of Cameroon and is located in the central-western part of the country at $5°30'N$ and $10°30'E$. It has an area of 13,892 $km^2$ sharing borders with the North West Region to the northwest, the Adamawa Region to the northeast, the Centre Region to the southeast, the Littoral Region to the southwest, and the South West Region to the west. This Region is one of the economic hotspots of the country, having the smallest area yet the second highest population density; a population of 1,865,394 (2013) with a density of 142.9 inhabitants/$km^2$ as of 2017 [15, 16]. Its capital is Bafoussam, in the Mifi Division. It is divided administratively into 8 Divisions: Bamboutos, Upper-Nkam, Upper-Plateau, Koung-Khi, Ménoua, Mifi, Ndé, and Noun [16]. The Region lies in the Grass Field plateaus of the Western Highlands with a cold climate. The major ethnic groups are the Bameliké and the Bamoum [17]. Inhabitants are greatly involved in animal husbandry, mainly poultry and pig farming, which serves as a main source of meat for households and mass catering events in the Region known for its attachment to cultural celebrations, including burial and funeral celebrations [18].

Isolates used in this study were from a previous study that investigated the prevalence and risk factors of *Enterobacteriaceae* carriage as well as resistance and resistance genes in these bacteria isolated from broiler chicken in poultry farms across the West Region of Cameroon. Sampling was stratified, with administrative Divisions as strata within which farms were selected. Poultry farms are controlled by the Ministry of Livestock, Fisheries and Animal Industries (MINEPIA in its French acronym), thus an authorization, reference number 68/18/L/DREPIA-O/SRAG of 04/06/2018 [S1 Appendix], to sample chicken within the West Region was obtained from the Regional Delegation of MINEPIA addressed to Divisional Delegates to enable collaboration with farmers. The consent of poultry farm owners to sample broilers was verbal after presentation of researcher credentials, research authorization, and an explanation of the work and the sampling procedure. The samples collected were cloacal swabs from healthy broilers.

### 2.2 Selection of isolates

*Enterobacteriaceae* organisms from the previous study mentioned above, belonging to the genera *E. coli*, *Salmonella* spp. (typhoidal serovars *S.* Typhi and *S.* Paratyphi A), and *Klebsiella pneumoniae*, were used for this study. 200 isolates were selected for detection of plasmid-borne resistance genes against quinolones and aminoglycosides and for extended spectrum beta lactamase (ESBL) production, including 100 resistant isolates and 100 susceptible isolates in *in-vitro* AST test by disc diffusion, and distributed as shown in Table 1 below.

**Table 1. Isolates used and their distribution.**

| Organism | | ESBLs | | Quinolones | | Aminoglycosides | |
|---|---|---|---|---|---|---|---|
| | | Positive | Negative | Resistant | Susceptible | Resistant | Susceptible |
| *E. coli* | | 10 | 10 | 14 | 14 | 10 | 10 |
| *Salmonella* | Typhi | 7 | 8 | 9 | 10 | 7 | 6 |
| | Paratyphi A | 3 | 2 | 5 | 4 | 3 | 4 |
| *K. pneumoniae* | | 10 | 10 | 12 | 12 | 10 | 10 |
| **TOTAL** | | **30** | **30** | **40** | **40** | **30** | **30** |

The antibiotics used in the AST were ceftriaxone, and cefotaxime, along with amoxicillin and amoxicillin/clavulanic acid for the preliminary selection of ESBL producers, levofloxacin and ciprofloxacin for the selection of quinolone-resistant isolates, and amikacin and gentamycin for the selection of aminoglycoside-resistant isolates. The EUCAST standard radii of inhibition for antibiotic susceptibility were used [19].

### 2.3 Antibiotic susceptibility testing

Among the selected susceptible isolates, those with a reduced diameter of inhibition, interpreted as having reduced susceptibility, were selected. To determine isolates that fell within this category (since no standard protocol describes reduced susceptibility for susceptible isolates with disk diffusion), the difference between the breakpoint value for susceptibility and the value of the largest diameter observed for a given antibiotic was calculated. This difference was divided by 4, and those isolates in the lower quadrant were selected. AST by broth microdilution was carried out on these selected isolates to determine an increase in the MIC of antibiotics to the isolates.

Antibiotic stock solutions were prepared from powdered stocks. A volume of DMSO equal to 5% of the total volume was added to the masses of pure antibiotics required for the various concentrations, and the final volume was completed with MHB. Bacterial suspensions were prepared at a concentration of $1.5 \times 10^8$ CFU/mL using a 0.5 McFarland scale from bacterial colonies aged 18 to 24 hours in 10 ml of sterile physiological water. The concentration was adjusted to obtain an inoculum solution of $2 \times 10^6$ CFU/mL by dilution with MHB. A volume of 100 µl of MHB was introduced into each well of a microplate with 96 wells. 100 µl of prepared antibiotic solutions were added, and the solution was serially diluted at a geometric progression of ratio 2 to establish a range of variable antibiotic concentrations in the 8 wells of each row. A volume of 100 µl of bacterial inoculum at $2 \times 10^6$ CFU/ml was introduced into each well for a final density of $10^6$ CFU/ml giving a final volume of 200 µl/well. The microplates were incubated at 37˚C for 18 hours. After incubation, bacterial growth was revealed in each well using 40 µl (per well) of a 0.2 mg/ml INT (para-iodonitrotetrazolium chloride) solution [20, 21]. The MICs were defined as the minimum concentrations for which bacterial growth was not observed (absence of pink coloration in the well). The tests were repeated three times and carried out in duplicate. Reduced susceptibility determined by broth dilution was interpreted as breakpoint values tending to the upper limit. The values of the inhibition zones for reduced susceptibility and MIC for *Enterobacteriaceae* considered are given in Table 2 below.

### 2.4 Detection of plasmid-borne resistance genes by PCR

Amplification of some representative plasmid-borne ESBL genes, plasmid mediated quinolone resistance (PMQR) genes and plasmid mediated aminoglycoside resistance (PMAR) genes in

**Table 2. Antibiotics used, their MIC, breakpoint zones of inhibition and range of reduced susceptibility.**

| Antibiotic | CTX | CRO | CTZ | CIP | LEV | CN | AMK |
|---|---|---|---|---|---|---|---|
| MIC (mg/L) for *Enterobacteriaceae* [22] | 0.03–128 | 0.001–128 | 0.004–128 | 0.004–128 | 0.004–128 | 0.03–128 | 0.03–128 |
| Breakpoint zones of inhibition (mm) [19] | 20 | 25 | 22 | 25 | 23 | 17 | 18 |
| Range of reduced susceptibility calculated (mm) | ≤25≥20 | ≤29≥25 | ≤28≥22 | ≤29≥25 | ≤27≥23 | ≤22≥17 | ≤22≥18 |

CTX = Cefotaxime, CRO = Ceftriaxone, CTZ = Ceftazidime, CIP = Ciprofloxacin, LEV = Levofloxacin, AMK = Amikacin CN = Gentamycin

both susceptible and resistant isolates was done by standard endpoint PCR. The following primer sets and reaction conditions in Table 3 were used.

DNA was extracted from fresh overnight colonies by heat shock. A loop-full of fresh colony was dissolved in 400 μl of Tris-EDTA 1X buffer (Tris-Cl 0.1 M and EDTA 0.01 M diluted 1/10), the solution was heated in a water bath at 95°C for 25 minutes, centrifuged at 13000 rpm for 3 minutes, and the supernatant containing DNA was extracted and used for PCR [27].

**Table 3. Primers used for the amplification of resistance genes.**

| Gene | Primer | Sequence (5'– 3') | Size of amplicon (bp) | Annealing temperature (°C) | Reference |
|---|---|---|---|---|---|
| | | ESBL genes | | | |
| *bla*TEM | BLATEM-F | ATAAAATTCTTGAAGACGAAA | 1080 | 53 | [23]. |
| | BLATEM-R | GACAGTTACCAATGCTTAATC | | | |
| *bla*TEM-1 | BLATEM-1-F | GGTCGCCGCATACACTATTC | 500 | 57 | |
| | BLATEM-1-R | ATACGGGAGGGCTTACCATC | | | |
| *bla*TEM-2 | BLATEM-2-F | AAGTAAAAGATGCTGAAGATAAGTTGG | 737 | 61 | |
| | BLATEM-2-R | GATCTGTCTATTTCGTTCATCCATAG | | | |
| *bla*CTX-M | BLACTX-M-F | GTGAAACGCAAAAGCAGCTG | 400 | 61 | |
| | BLACTX-M-R | CCGGTCGTATTGCCTTTGAG | | | |
| *bla*SHV-1 | BLASHV-1-F | GCGTTATATTCGCCTGTGTATTAT | 385 | 58 | |
| | BLASHV-1-R | GCCTGTTATCGCTCATGGTAATG | | | |
| *bla*KPC | BlaKPC F | TGTCACTGTATCGCCGTC | 900 | 58 | [24] |
| | BlaKPC R | CTCAGTGCTCTACAGAAACC | | | |
| | | PMQR genes | | | |
| *qnrA* | QNRA-F | TCAGCAAGAGGATTTCTCA | 627 | 58 | [25]. |
| | QNRA-R | GGCAGCACTATTACTCCCA | | | |
| *qnrB* | QNRB-F | GATCGTGAAAGCCAGAAAGG | 476 | 58 | |
| | QNRB-R | ACGATGCCTGGTAGTTGTCC | | | |
| *qnrS* | QNRS-F | ATGGAAACCTACAATCATAC | 491 | 58 | |
| | QNRS-R | AAAAAACACCTCGACTTAAGT | | | |
| *aac(6')IB-CR* | AAC(6')IB-CR-F | TTGCGATGCTCTATGAGTGGCTA | 482 | 58 | |
| | AAC(6')IB-CR-R | CTCGAATGCCTGGCGTGTTT | | | |
| *qepA* | QEPA-F | GCAGGTCCAGCAGCGGGTAG | 199 | 60 | |
| | QEPA-R | CTTCCTGCCCGAGTATCGTG | | | |
| | | PMAR genes | | | |
| *aac(6')-IB* | aac(6')-Ib–F | AGTACTTGCCAAGCGTTTTAGCGC | 365 | 58 | [26]. |
| | aac(6')-Ib–R | CATGTACACGGCTGGACCAT | | | |
| *aph(3')-IA* | aph(3')-Ia–F | ATGGGCTCGCGATAATGTCG | 734 | 57 | |
| | aph(3')-Ia–R | AGAAAAACTCATCGAGCATC | | | |
| *ant(2')-IA* | ant(2')-Ia–F | ATGCAAGTAGCGTATGCGCT | 477 | 57 | |
| | ant(2')-Ia–R | TCCCCGATCTCCGCTAAGAA | | | |

**Table 4. Frequency of reduced susceptibility observed in susceptible isolates.**

| Organism | | ESBLs n/N (%) | Quinolones n/N (%) | Aminoglycosides n/N (%) |
|---|---|---|---|---|
| *E. coli* | | 4/10 (40.00) | 6/14 (42.86) | 6/10 (60.00) |
| *Salmonella* | Typhi | 3/8 (37.50) | 7/10 (70.00) | 4/6 (66.67) |
| | Paratyphi A | 0/2 (0.00) | 2/4 (50.00) | 3/4 (75.00) |
| *K. pneumoniae* | | 2/10 (20.00) | 5/12 (41.67) | 3/10 (30.00) |
| **Total** | | **9/30 (30.00)** | **20/40 (50.00)** | **16/30 (53.33)** |

Where N = total number of susceptible isolates selected, n = number of isolates showing reduced susceptibility.

PCR was done with a 25 μl reaction mix composed of 14.9 μL of PCR grade water, 2.5 μL of 1X standard Taq buffer solution with 2.5 mM $MgCl_2$ (1X), 1 μL of forward primer (0.4 μM), 1 uL of reverse primer (0.4 μM), 0.5 μL of DNTP mix (200 μM), 0.1 μL of standard Taq (0.2 U/μl) and 5 μL of DNA solution in a TECHNE[®] thermocycler [28]. Reaction products were migrated on a 1.5% agarose gel and revealed under UV light.

Arithmetic operations and conversions were done using Microsoft Excel 2016 calculation sheets, while statistical analysis was done using IBM SPSS Statistics 20.

## 3 Results

### 3.1 Reduced antibiotic susceptibility profiles

Among the 100 susceptible isolates, 45 were assessed to have reduced susceptibility, which was confirmed by broth microdilution. These were distributed as given in Table 4. The frequency of reduced susceptibility is presented first for the respective genera and then overall for all the organisms.

In total, among the susceptible isolates selected for testing of resistance genes, 30% had reduced susceptibility to cephalosporins (ceftriaxone and/or cefotaxime), 50% had reduced susceptibility to quinolones (ciprofloxacin and/or levofloxacin), and 53% had reduced susceptibility to aminoglycosides (gentamicin and/or amikacin).

### 3.2. Plasmid-borne resistance genes detected

**3.2.1 Resistance genes in susceptible isolates versus reduced susceptibility.** Plasmid-borne resistance genes against the 3 antibiotic classes tested were detected in phenotypically susceptible isolates and in all three genera, with overall prevalences of 60.00%, 65.00% and 66.67% for ESBL production, PMQRs, and PMARs, respectively. Below in Table 5, the cumulated prevalence of resistance genes for the antibiotic classes tested is presented. Counted are

**Table 5. Prevalence of resistance genes in susceptible isolates.**

| Organism | | ESBL genes n/N (%) | PMQR genes n/N (%) | PMAR genes n/N (%) |
|---|---|---|---|---|
| *E. coli* | | 6/10 (60.00) | 10/14 (71.43) | 7/10 (70.00) |
| *Salmonella* | Typhi | 4/8 (50.00) | 7/10 (70.00) | 4/6 (66.67) |
| | Paratyphi A | 0/2 (0.00) | 2/4 (50.00) | 4/4 (100.00) |
| *K. pneumoniae* | | 5/10 (50.00) | 7/12 (58.33) | 5/10 (50.00) |
| **Total** | | **15/30 (50.00)** | **26/40 (65.00)** | **20/30 (66.67)** |

Where N = total number of isolates tested, n = number of isolates positive for given genes.

**Table 6. Association between the possession of resistance genes and reduced susceptibility.**

| Factor | Outcome | Risk estimate | | | Correlation (*significant correlation $\leq$ 0.05*) |
|---|---|---|---|---|---|
| | | Odds ratios (*increased risk > 1*) | 95% confidence interval | | |
| | | | Lower | Upper | |
| Reduced susceptibility | ESBL genes | 4.125 | 0.883 | 19.273 | 0.069 |
| | PMQR genes | 8.36 | 1.971 | 35.461 | 0.002 |
| | PMAR genes | 8.889 | 1.294 | 61.058 | 0.015 |

susceptible isolates with at least one resistance gene in each class. The prevalence rates are given in brackets as percentages.

The association between the possession of resistance genes and reduced susceptibility was analysed as shown in Table 6.

Analysis of odds ratios showed that reduced susceptibility increased the risk of susceptible isolates carrying resistance genes. Gene carriage correlated significantly with reduced susceptibility only for the PMQR and PMAR genes.

**3.2.2 Assessment of the prevalence of resistance genes isolated from resistant versus susceptible isolates.** Resistance genes against the three antibiotic classes detected in susceptible and resistant isolates are presented in Table 7.

There was a variable prevalence of the individual genes, with proportionate prevalence in resistant and susceptible isolates for a given gene. The details of the prevalence of each gene in the susceptible and resistant isolates are given in Table 8 below.

The two proportion z-score shows that there were significant differences in the proportions of the genes in the resistant and susceptible groups. A paired sample t-test performed on the prevalence as shown in Table 8 above showed a correlation between the two groups; resistant and susceptible, and a significant difference between the prevalence of resistance genes in resistant and susceptible isolates, as shown in Table 9 below.

The t score shows that the two groups are different. The p value shows that the detection of resistance genes in susceptible isolates did not occur by chance (manipulation error) but is a well-established situation.

## 3.3 Penetrance of the genes

Resistance genes show varied expressivity, thus the results of AST can be interpreted as resistant, intermediate, or susceptible in disc diffusion [12]. However, we can estimate the extent to which the bacteria carrying the resistance genes actually showed the threshold of gene expression to be phenotypically resistant through the calculation of the penetrance. Mathematically, penetrance is the ratio of the individuals showing the phenotype to the total individuals having the genotype, expressed as a percentage [29]. To estimate the penetrance of the plasmid-borne genes, we eliminated the individuals with phenotypic resistance but not carrying the plasmid-borne genes on the basis that the resistance would have been due to chromosomally encoded

**Table 7. Prevalence of resistance genes in resistant and susceptible isolates.**

| | Prevalence in resistant isolates n/N (%) | Prevalence in susceptible isolates n/N (%) |
|---|---|---|
| ESBL genes | 27/30 (90.00) | 15/30 (50.00) |
| PMQR genes | 33/40 (82.50) | 26/40 (65.00) |
| PMAR genes | 28/30 (93.33) | 20/30 (66.67) |

Where N = total number of isolates tested, n = number of isolates positive for the genes.

**Table 8. Detailed prevalence of the individual resistance genes in resistant and susceptible isolates.**

| Gene category | N | Gene | Prevalence in resistant isolates [n (%)] | Prevalence in susceptible isolates [n (%)] | two proportion z-score (significant difference ≤ 0.05) |
|---|---|---|---|---|---|
| **ESBL genes** | 30 | $bla_{TEM}$ | 8 (26.67) | 5 (16.67) | ≤ 0.0001 |
| | | $bla_{TEM-1}$ | 16 (53.33) | 9 (30.00) | ≤ 0.0001 |
| | | $bla_{TEM-2}$ | 5 (16.67) | 3 (10.00) | ≤ 0.0001 |
| | | $bla_{CTX-M}$ | 7 (23.33) | 0 (0.00) | ≤ 0.0001 |
| | | $bla_{SHV-1}$ | 2 (6.67) | 0 (0.00) | ≤ 0.0001 |
| | | $bla_{KPC}$ | 10 (33.33) | 7 (23.33) | ≤ 0.0001 |
| **PMQR genes** | 40 | qnrA | 15 (37.50) | 8 (20.00) | ≤ 0.0001 |
| | | qnrB | 11 (27.50) | 6 (15.00) | ≤ 0.0001 |
| | | qnrS | 21 (52.50) | 17 (42.50) | ≤ 0.0001 |
| | | qepA | 11 (27.50) | 8 (20.00) | ≤ 0.0001 |
| | | aac(6')-IB-CR | 22 (55.00) | 15 (37.50) | ≤ 0.0001 |
| **PMAR genes** | 30 | aac(6')-IB | 28 (93.33) | 24 (80.00) | ≤ 0.0001 |
| | | aph(3´)-IA | 12 (40.00) | 9 (30.00) | ≤ 0.0001 |
| | | ant(2´)-IA | 11 (36.67) | 6 (20.00) | ≤ 0.0001 |

Where N = total number of isolates tested, n = number of isolates positive for a given gene and the prevalence calculated as n/N expressed as percentage.

genes or plasmid-borne genes that were not considered in this study. The computed estimates of the penetrance of the genes as per the interpretations of AST are shown in Table 10.

$$\text{Penetrance} = \frac{\text{phenotypically resistant and having the resistance gene}}{(\text{phenotypically resistant and having the resistance gene} + \text{phenotypically susceptible but having the resistance gene}} \times 100$$

This shows that 64.28% of isolates possessing the ESBL genes actually showed the ESBL production phenotype in AST by disc diffusion, 56.90% of isolates possessing PMQR genes, and 58.33% of isolates with PMAR genes actually showed the resistance phenotype in AST by disc diffusion.

To determine how, in our study, the result of phenotypic AST could indicate the presence of plasmid-borne resistance, we calculated the diagnostic odds ratio of a positive phenotypic

**Table 9. Paired samples correlations for prevalence of resistance genes in resistant and susceptible isolates.**

| **Paired samples correlations** | | | | |
|---|---|---|---|---|
| | | N | Correlation | Significance at CI$_{95}$ |
| 1 pair | Prevalence in resistant isolates & Prevalence in susceptible isolates | 14 | 0.964 | ≤0.0001 |
| **Paired samples test** | | | | |
| | | Paired differences | T | Significance at (2-tailed) CI$_{95}$ |
| | | 95% CI of the difference | | |
| | | Lower \| Upper | | |
| 1 pair | Prevalence in resistant isolates—Prevalence in susceptible isolates | 3.326 \| 5.531 | 8.675 | ≤0.0001 |

**Table 10. Penetrance of the genes.**

| | Resistant isolates | | Susceptible isolates | | Penetrance (%) |
|---|---|---|---|---|---|
| | Total | Positive for gene | Total | Positive for gene | |
| ESBL genes | 30 | 27 | 30 | 15 | $\frac{27}{27+15}$ x 100 = 64.28 |
| PMQR genes | 40 | 33 | 40 | 25 | $\frac{33}{33+25}$ x 100 = 56.90 |
| PMAR genes | 30 | 28 | 30 | 20 | $\frac{28}{28+20}$ x 100 = 58.33 |

**Table 11. Diagnostic odds ratio of a positive phenotypic resistance.**

| Resistance | phenotypically resistant and having the resistance gene (TP) | phenotypically resistant but not having the resistance gene (FP) | Reliability of positive phenotypic resistance in the search for resistance gene (diagnostic odds ratio; DOR) [30] |
|---|---|---|---|
| | phenotypically susceptible but having the resistance gene (FN) | phenotypically susceptible and not having the resistance gene (TN) | $DOR = \frac{TP x TN}{FP x FN}$ |
| ESBL | TP = 27 | FP = 3 | 9.00 |
| | FN = 15 | TN = 15 | |
| PMQR | TP = 33 | FP = 7 | 2.83 |
| | FN = 25 | TN = 15 | |
| PMAR | TP = 28 | FP = 2 | 3.50 |
| | FN = 24 | TN = 6 | |

resistance as shown in Table 11. A value greater than one shows that phenotypic resistance is effective in detecting isolates with resistance genes.

## 4 Discussion

The aim of this study was to determine whether phenotypic susceptibility directly translates into the absence of resistance genes or determinants and to demonstrate if resistance determinants can build up in susceptible isolates and will go unchecked when routine procedures target only resistant isolates.

At the end of this study, we observed that phenotypically susceptible isolates carried plasmid-borne resistance genes against quinolones, aminoglycosides, and for the production of beta lactamase at 62.50%, 66.67% and 50.00% respectively. All three genera: *E. coli*, *Salmonella* spp. and *Klebsiella pneumoniae* carried unexpressed genes, showing that this phenomenon is ubiquitous among them. The presence of resistance genes in susceptible isolates shows that an organism can carry a resistance gene against a given AMA whose expression is either reduced or hampered by several factors, both genetic and epigenetic.

From a genetic engineering standpoint, the expression of foreign proteins in a host cell can be influenced by a variety of factors, among which are promoter strength, efficiency of ribosome binding, stability of the foreign protein in the host cell, the metabolic state of the cell, and the stability and copy number of the foreign gene [31]. Deekshit and collaborators suggested that deletion in the promoter region was the principal reason behind the unexpressive nature of the chloramphenicol acetyltransferase (*catA*) gene in *Salmonella Weltevreden* [32]. Under stress conditions such as antibiotic treatment, several genes, including innate mechanisms of defence against antibiotics, can have expression that is critically nonoptimal, such that the genetic content does not always match the expression (phenotype) [33]. Also, antibiotic resistance is an extra function with an additional biological cost that bacteria tend to overcome by several means, including expression only upon induction by an antibiotic, which may be dose-dependent and leads to poor expression at concentrations of the wild type's MIC [34].

Another reason for reduced susceptibility can be "low level" mutations at the target site of the antibiotic; for example *E. coli* with a single mutation in *gyrA* remains susceptible to fluoroquinolones but with significantly increased MICs [35]. A gene in the resistome that does not have the capacity to induce stronger levels of antibiotic resistance will only be detected, if at all, phenotypically by hypersensitivity screening with drug concentrations below the wild type's MIC. Thus, screening for resistance with drug concentrations above the MIC will only identify organisms with optimal expression and gene overexpression mutants. Thus, there is no efficient and sensitive screening method for nonoptimally expressed genes [33]. These unexpressed genes, especially those carried on MGEs, would continue to be disseminated, especially in a luxuriant microbial community like the gut microbiota [36].

The existence of resistance genes in susceptible isolates correlated significantly with reduced susceptibility (p = 0.002 for PMQR genes and p = 0.015 for PMAR). This shows a lower level of expression not detected using standard thresholds. When ABR is polygenic, a high level of resistance can be attained by what Cavalli and Maccacaro termed "training", in which successive exposure to increasing doses of an antibiotic increases the resistance level in bacteria, which they showed to attain up to 200 times the initial susceptible dose [37]. These isolates might not have yet had sufficient exposure to antibiotics to mobilise their resistome. We see here that AST grouping organisms as resistant or susceptible in a clinical setting fully serves the purpose of AST, which is to select the best AMA to treat a disease. However, in epidemiological studies, this will hide resistance genes and resistance patterns that are gradually being activated and enriched in the community.

Despite the prevalence of the resistance genes in susceptible isolates, the higher prevalence of these very genes in resistant isolates, which statistically is different from the former (t value of 8.675 and a p value of $\leq$ 0.0001), coupled with the association of reduced susceptibility with the resistance gene in susceptible isolates (OR 4.125, p value 0.069 for ESBLs; OR 8.36, p value 0.002 for PMQR genes; and OR 8.89, p value 0.015 for PMAR genes) indicate that these resistance genes are responsible for or contribute to the overall phenotype (the resistance). The prevalence of the respective genes in both resistant and susceptible isolates was proportionate, indicating a uniform overall prevalence of each gene in the community.

In estimating the extent to which gene carriage corresponded to the phenotypic resistance defined as resistant and susceptible in AST, we considered the notion of penetrance, making two considerations: we excluded the resistant isolates that were not positive for the genes, assuming that the resistance was due to other mechanisms or plasmid-borne genes not considered in this study. We also assumed that the observed resistance in resistant isolates that tested positive for the genes was effectively due to the genes. The estimation showed that these genes had the following penetrances: ESBL genes (64.28%), PMQR genes (56.90%), and PMAR genes (58.33%). This shows the extent to which plasmid-borne genes can be silent in these bacteria, with the risk of mobilization by "training" [37] in the face of continuous antibiotic abuse. Several reports show a similar situation where various *Enterobacteriaceae* organisms carry unexpressed or underexpressed genes [4, 38, 39]. The genes *blaTEM1*, *blaKPC*, *qnrA*, *qnrB*, *aac(6')-Ib-cr*, and *aac(6')-Ib* showed a high prevalence in susceptible isolates. These are clinically and epidemiologically important genes with a serious threat to humans due to their horizontal mobility and high enrichment in human-associated environments and pathogens [6]. This calls for concern because these genes may be further enriched in the community and become activated or have their expression optimised. Plasmid-borne genes are acquired by horizontal gene transfer. New genes acquired by a cell are mobilised for expression, or their expression is optimised based on several factors. The gene has to integrate the gene regulation networks of the new host for its appropriate expression. One main way this happens is by the opposing mechanisms of xenogeneic silencing and counter-silencing, which favour ancestral

gene expression over acquired genes, which are only expressed if they increase the organism's fitness. Acquired genes are expressed easily when their expression contributes to the fitness of the organism and does not act as an extra cost to the organism. In the case of the unexpressed or underexpressed genes, we can postulate that these genes have just been newly acquired by the organism and have remained silent because there has been insufficient prior exposure to antibiotics for the cell to mobilise them into use or optimise their expression. Thus, the low penetrances observed suggest recent and extensive dissemination of the plasmids bearing these genes [34, 37, 40–42].

Analysis of the diagnostic odds ratio of a positive phenotypic resistance (9.00 for ESBLs, 2.83 for PMQR genes, and 3.50 for PMAR genes) shows that in selecting isolates with potential resistance determinants, screening can still be effectively done by selecting the phenotypically resistant isolates. However, as seen in this study, susceptible isolates can carry the resistance genes and go undetected.

This research only demonstrated that susceptible isolates could carry resistance genes but did not point out the reason for the non-expression or suboptimal expression of these genes. Further research will be carried out to identify, among the options given in this discussion, the mechanism responsible for the nonoptimal or non-expression of the genes.

## 5 Conclusion

We report a high carriage of plasmid mediated genes for ESBL, quinolone, and aminoglycoside resistance in *Enterobacteriaceae* isolates from broilers in West Region of Cameroon. However, the presence of these genes was not always correlated with phenotypic expression, allowing us to hold that prior AST for selection of resistant isolates in epidemiological studies will exclude susceptible isolates carrying resistance gene, and underestimate the potential of silent dissemination of resistance within a microbial community. Other mechanisms such as chromosomal modulatory mechanisms and antibiotic-induced bacteria fitness, could contribute to maintain the viability of bacteria despite a lethal antibiotic dose.

## Supporting information

**S1 Appendix. Research authorization.**
(PDF)

## Acknowledgments

We wish to thank the Research Unit of Microbiology and Antimicrobial Substances (RUMAS) that provided the work space. We equally thank Pr. Simo Gustave of the Molecular Parasitology and Entomology Subunit (MPES) for the technical platform and coaching offered during the molecular analysis phase of the work. We thank the poultry farmers who accepted to participate in this research by allowing sampling among their broilers.

## Author Contributions

**Conceptualization:** Innocent Mbulli Ali, Christopher B. Tume.

**Data curation:** Jude Fonbah Leinyuy.

**Formal analysis:** Jude Fonbah Leinyuy, Karimo Ousenu.

**Investigation:** Jude Fonbah Leinyuy, Karimo Ousenu.

**Methodology:** Jude Fonbah Leinyuy, Innocent Mbulli Ali, Karimo Ousenu, Christopher B. Tume.

**Resources:** Jude Fonbah Leinyuy, Innocent Mbulli Ali.

**Supervision:** Christopher B. Tume.

**Validation:** Innocent Mbulli Ali, Christopher B. Tume.

**Writing – original draft:** Jude Fonbah Leinyuy.

**Writing – review & editing:** Innocent Mbulli Ali, Karimo Ousenu, Christopher B. Tume.

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
