## [Decision Letter · Decision Letter 0]

3 Oct 2022

PONE-D-22-06329Characterisation of antimicrobial resistance genes and assessment of penetrance in plasmid-mediated resistance genes in E coli, Salmonella and Klebsiella in the West Region of Cameroon.PLOS ONE

Dear Dr. Ali,

Thank you for submitting your manuscript to PLOS ONE. After careful consideration, we feel that it has merit but does not fully meet PLOS ONE’s publication criteria as it currently stands. Therefore, we invite you to submit a revised version of the manuscript that addresses the points raised during the review process.

We look forward to receiving your revised manuscript.

Kind regards,

Dwij Raj Bhatta, PhD

Academic Editor

PLOS ONE

Journal Requirements:

**When submitting your revision, we need you to address these additional requirements.**

**1. Please ensure that your manuscript meets PLOS ONE's style requirements, including those for file naming. The PLOS ONE style templates can be found at **

**https://journals.plos.org/plosone/s/file?id=wjVg/PLOSOne_formatting_sample_main_body.pdf and **

**
https://journals.plos.org/plosone/s/file?id=ba62/PLOSOne_formatting_sample_title_authors_affiliations.pdf
**

3. We note you have included a table to which you do not refer in the text of your manuscript. Please ensure that you refer to Table 1,2,5,6,7,9,10 & 11 in your text; if accepted, production will need this reference to link the reader to the Table

4. We suggest you thoroughly copyedit your manuscript for language usage, spelling, and grammar. If you do not know anyone who can help you do this, you may wish to consider employing a professional scientific editing service.

Additional Editor Comments (if provided):

Country specific publication on circulating antibiotic resistant isolates of E.coli,Klebsiella Salmonella and molecular charecterization of resistance related genes has been reviewed and the manuscript require minor revision as suggested by reviewers!

Answer all queries from reviewer! Revise as suggested!

TItle change suggested may be "Molecular charecterization of AMR related genes in E.coli,Klebsiella and Salmonella isolates from Cameroon

Reviewers' comments:

Reviewer's Responses to Questions

**Comments to the Author**

1. Is the manuscript technically sound, and do the data support the conclusions?

Reviewer #1: Yes

2. Has the statistical analysis been performed appropriately and rigorously? 

Reviewer #1: Yes

3. Have the authors made all data underlying the findings in their manuscript fully available?

Reviewer #1: Yes

4. Is the manuscript presented in an intelligible fashion and written in standard English?

Reviewer #1: Yes

5. Review Comments to the Author

Reviewer #1: This study highlights the importance of molecular methods in studying the dissemination of a particular resistance determinant in both resistant and susceptible isolates. I have only minor comments

Major comments:

1.In the Financial disclosure section, the authors mentioned that “The funders had no role in study design, data collection and analysis, decision to publish, or preparation of the manuscript”. Please provide the details related to the fund as per the author’s guideline.

2.In ethics statement section, the authors need to mention the approval number, type of consent –oral or written, etc.

3.In abstract, the methods part is missing. The main methods should be mentioned in the abstract section.

Minor comments:

Please correct the grammatical and typing errors in the text such as

1.In introduction section, last paragraph:

“We focused on 3 genera o……………………….. which known for the extensive colonisation of their guts by a myriad of Enterobacteriaceae”

Please rearrange the sentence

1.Please italicize the genus and species of bacteria throughout the text. For eg, in methods section, E.coli should be italicized.

6. PLOS authors have the option to publish the peer review history of their article (what does this mean?). If published, this will include your full peer review and any attached files.

Reviewer #1: No

---

## [Author Response · Author response to Decision Letter 0]

21 Oct 2022

Reviewer comments

This study highlights the importance of molecular methods in studying the dissemination of a particular resistance determinant in both resistant and susceptible isolates. I have only minor comments

Major comments:

In the Financial disclosure section, the authors mentioned that “The funders had no role in study design, data collection and analysis, decision to publish, or preparation of the manuscript”. Please provide the details related to the fund as per the author’s guideline.

The authors declared at submission while requesting for country-based waiver of publication fees, that the research was not externally funded. Resources for the research were provided by the authors. So we attached the letter from the Head of Department to that effect.

The financial disclosure statement selected might have been a submission error which has now been corrected in the resubmission.

In ethics statement section, the authors need to mention the approval number, type of consent –oral or written, etc. 

In Cameroon, the only well-established institutional review board deals with ethical clearance of research on human subjects (the National Ethical Committee for Research involving Human Subjects – CNERSH/SP – Yaoundé, Cameroon). However, at the level of the universities, research projects dealing with animals are reviewed by experts and the protocol is submitted for administrative authorisation from the Ministry of Livestock, Fisheries and Animal Husbandry with a recommendation from the Head of Department concerned.

 In addition, Law number 2021/014 of 09/07/2021 regulating access and use of genetic material and its derivatives in the Cameroonian territory in its article 4 is not binding to biological resources whose usage is not aimed at exploiting the genetic resources, as the case with our research (https://www.prc.cm/en/news/the-acts/laws/5293).

A statement on the consent from farmers to sample broilers at farms has been included in the method section lines 101 to 104. 

In abstract, the methods part is missing. The main methods should be mentioned in the abstract section.

The methods part in the Abstract has been updated to be explicit from lines 19 to 25.

Minor comments:

Please correct the grammatical and typing errors in the text such as 

In introduction section, last paragraph:

“We focused on 3 genera o……………………….. which known for the extensive colonisation of their guts by a myriad of Enterobacteriaceae”

The sentence has been rearranged to bring out the intended meaning, see lines 71 to 74.

Please italicize the genus and species of bacteria throughout the text. For eg, in methods section, E.coli should be italicized.

The manuscript has been thoroughly reviewed and scientific names italicised.

Additional remarks

The manuscript has been thoroughly updated to meet journal’s specifications

We refer the Reviewer to previous comments in response to this concern (number 2 above)

3. We note you have included a table to which you do not refer in the text of your manuscript. Please ensure that you refer to Table 1,2,5,6,7,9,10 & 11 in your text; if accepted, production will need this reference to link the reader to the Table

We have made sure In-text table citations have been included as recommended

4. We suggest you thoroughly copyedit your manuscript for language usage, spelling, and grammar. If you do not know anyone who can help you do this, you may wish to consider employing a professional scientific editing service.

The manuscript has been thoroughly proofread and language usage, spelling, and grammar errors corrected 

5. Please review your reference list to ensure that it is complete and correct.

The references have been re-checked to ensure that they match the journal’s specifications 

Editor recommendation for title change

Title change suggested may be "Molecular characterization of AMR related genes in E. coli, Klebsiella and Salmonella isolates from Cameroon”

The authors feel that changing the title may in part mask the main aim of the research which not only pointed to the problem of resistance but raises the hypothesis of a mutated (resistance) gene (which does not necessarily translate to being resistant as detected in in-vitro AST) to contribute to the fate of the isolate (penetrance) and hence the ability to silently circulate in the study area.

---

## [Editor Report · Decision Letter 1]

15 Nov 2022

PONE-D-22-06329R1Characterisation and assessment of penetrance in plasmid-mediated antimicrobial resistance genes in E coli, Salmonella and Klebsiella in the West Region of Cameroon.PLOS ONE

Dear Dr. Ali,

Thank you for submitting your manuscript to PLOS ONE. After careful consideration, we feel that it has merit but does not fully meet PLOS ONE’s publication criteria as it currently stands. Therefore, we invite you to submit a revised version of the manuscript that addresses the points raised during the review process.

We look forward to receiving your revised manuscript.

Kind regards,

Dwij Raj Bhatta, PhD

Academic Editor

PLOS ONE

Journal Requirements:

Additional Editor Comments:

Revision 1 manuscript has been reviewed by editor :Abstrct section ,methodology, poorly written, recheck spelling, simplify language, cut shot abstract section! In result section, catagorize,clearly mention number of sucestible isolates by disc diffusion method and minimum inhibitory concentration of each antibiotics used during susceptibility detetmination! Mention name of Salmonella serovars ,was all Typhi?plasmid en

coded AMR genes in Salmonella ,E.coli,and Klebsiella are not charectrized with propoer protocol! Revise result presentation! conclusion : poorly written provide result based conclusion

Change title as sugested previously by editor , as editor is not convicend with conclusion of penetrance! remove the section

---

## [Author Response · Author response to Decision Letter 1]

7 Dec 2022

Response to Editor Comments

Abstract section, methodology, poorly written, recheck spelling, simplify language, cut short abstract section! 

The methodology in the abstract has now been rewritten clearly, however much could not be done to contract the section without leaving out essential information. Lines 21 – 27.

In result section, categorize, clearly mention number of susceptible isolates by disc diffusion method and minimum inhibitory concentration of each antibiotic used during susceptibility determination! 

The result section has been clearly categorised and the presentation of the tables simplified. In addition, The MIC of the antibiotics used can be found in line 151, Table 1.

Mention name of Salmonella serovars, was all Typhi? 

The Salmonella serovars used have been specified. Line 112.

Plasmid encoded AMR genes in Salmonella, E. coli, and Klebsiella are not characterized with proper protocol! 

Authors noted some commissions in the protocol and thank the reviewer for bringing this to our attention. We have now corrected this, mentioning clearly the necessary conditions, quantities and concentrations required for independent reproducibility of the resistance genes characterisation. Lines 161 – 171.

Revise result presentation! 

The result section has been thoroughly revised.

Conclusion: poorly written provide result based conclusion

The conclusions have been reworded to reflect the primary findings of this work. This can be found in Lines 341 – 348. 

Change title as suggested previously by editor, as editor is not convinced with conclusion of penetrance! remove the section.

The title has been modified to “Molecular characterization of antimicrobial resistance related genes in E. coli, Salmonella and Klebsiella isolates from the West Region of Cameroon” as suggested by the Editor.

 Authors used the notion of penetrance here to made adequacy between genotype and phenotype. Though the expression of antibiotic resistance can vary in intensity depending on prevailing conditions, we used the notion of penetrance by extrapolation based the interpretations of disc diffusion AST only to give an idea of the proportion of isolates considered susceptible that will carry resistance genes.

---

## [Editor Report · Decision Letter 2]

12 Dec 2022

PONE-D-22-06329R2Molecular characterization of antimicrobial resistance related genes in E. coli, Salmonella and Klebsiella isolates from the West Region of Cameroon.PLOS ONE

Dear Dr. Ali,

Thank you for submitting your manuscript to PLOS ONE. After careful consideration, we feel that it has merit but does not fully meet PLOS ONE’s publication criteria as it currently stands. Therefore, we invite you to submit a revised version of the manuscript that addresses the points raised during the review process.

We look forward to receiving your revised manuscript.

Kind regards,

Dwij Raj Bhatta, PhD

Academic Editor

PLOS ONE

Journal Requirements:

Additional Editor Comments :

Revision 2 manuscript is well written! However,

in the abstract section word treat be replaced

by word threat! Salmonella serovars names either S.Tphi or S.ParatyphiA or S.Paratyphi B or S.Paratyphi C and their sucesptibility

to tested antibiotics

be presented separately not as Salmonella spp.

---

## [Author Response · Author response to Decision Letter 2]

20 Dec 2022

Response to reviewers 

Journal Requirements

• The financial disclosure provided remains unchanged.

• If applicable, we recommend that you deposit your laboratory protocols in protocols.io to enhance the reproducibility of your results. Protocols.io assigns your protocol its own identifier (DOI) so that it can be cited independently in the future. 

No special protocol was developed by the authors in this manuscript.

• Please review your reference list to ensure that it is complete and correct. If you have cited papers that have been retracted, please include the rationale for doing so in the manuscript text, or remove these references and replace them with relevant current references. 

The reference list has been thoroughly reviewed and none of the articles, web pages, and videos cited has been retracted. No modification has been made on the references.

• While revising your submission, please upload your figure files to the Preflight Analysis and Conversion Engine (PACE) digital diagnostic tool, https://pacev2.apexcovantage.com/. PACE helps ensure that figures meet PLOS requirements. 

The manuscript does not contain images.

Additional Editor Comments 

• In the abstract section word treat be replaced by word threat! 

We thank the Editor for pointing out this spelling error which escaped out notice despite several revisions. Line 14. To fish out such errors, the manuscript has been submitted to a spelling and grammar check web site (https://quillbot.com) and necessary corrections have been made. 

• Salmonella serovars names either S.Tphi or S.ParatyphiA or S.Paratyphi B or S.Paratyphi C and their sucesptibility to tested antibiotics be presented separately not as Salmonella spp.

We have revised the research data to comply with the revision requested. Lines 113, 119, 181, and 196.

---

## [Editor Report · Decision Letter 3]

21 Dec 2022

Molecular characterization of antimicrobial resistance related genes in E. coli, Salmonella and Klebsiella isolates from broilers in the West Region of Cameroon.

PONE-D-22-06329R3

Dear Dr. Ali,

We’re pleased to inform you that your manuscript has been judged scientifically suitable for publication and will be formally accepted for publication once it meets all outstanding technical requirements.

Kind regards,

Dwij Raj Bhatta, PhD

Academic Editor

PLOS ONE

Additional Editor Comments (optional):

The revision 3 manuscript be accepted for publication as it provides important information on AST pattern of emerging &circulating multidrug resistant E.coli , Klebsiella and Salmonella serovars in Camroon!
---

## [Editor Report · Acceptance letter]

29 Dec 2022

PONE-D-22-06329R3 

Molecular characterization of antimicrobial resistance related genes in *E. coli, Salmonella* and *Klebsiella* isolates from broilers in the West Region of Cameroon. 

Dear Dr. Ali:

I'm pleased to inform you that your manuscript has been deemed suitable for publication in PLOS ONE. Congratulations! Your manuscript is now with our production department. 

Kind regards, 

on behalf of

Professor Dwij Raj Bhatta 

Academic Editor

PLOS ONE